# Camellia Tea Saponin Ameliorates 5–Fluorouracil-Induced Damage of HaCaT Cells by Regulating Ferroptosis and Inflammation

**DOI:** 10.3390/nu17050764

**Published:** 2025-02-21

**Authors:** Tanrada Likitsatian, Pimpisid Koonyosying, Narisara Paradee, Sittiruk Roytrakul, Haobo Ge, Charareh Pourzand, Somdet Srichairatanakool

**Affiliations:** 1Department of Biochemistry, Faculty of Medicine, Chiang Mai University, Chiang Mai 50200, Thailand; tanrada.likit@gmail.com (T.L.); pimpisid.k@cmu.ac.th (P.K.); narisara.p@cmu.ac.th (N.P.); 2National Center for Genetic Engineering and Biotechnology (BIOTEC), National Science and Technology Development Agency, Khlong Luang 12120, Thailand; sittiruk@biotec.or.th; 3Department of Life Sciences, University of Bath, Bath BA2 7AY, UK; hg220@bath.ac.uk (H.G.); c.a.pourzand@bath.ac.uk (C.P.)

**Keywords:** 5–fluorouracil, ferroptosis, tea, saponin, oxidative stress, lipid peroxidation and inflammation

## Abstract

Background/Objective: Ferroptosis is an iron-dependent form of programmed cell death characterized by lipid peroxidation products (LPOs). A chemotherapeutic drug, 5–fluorouracil (5–FU), can induce epithelial mucositis and favor drug synergism with erastin in ferroptosis. *Camellia* tea saponin extract (TS) is known to exert antioxidative properties. This study aims to delineate the protective role of TS in mitigating 5–FU-induced ferroptosis and inflammation in human keratinocytes. Methods: HaCaT cells were induced by 5–FU and erastin, treated with different TS doses, and their viability was then determined. Levels of cellular reactive oxygen species (ROS), LPOs, labile iron pool (LIP), glutathione (GSH), glutathione peroxidase 4 (GPX–4) activity, as well as IL–6, IL–1β, and TNF–α levels, and their wound healing properties were assessed. Results: TS per se (at up to 25 µg/mL) was not toxic to HaCaT cells but was unable to restore the viability of 5–FU-induced cells up to the baseline levels. The compound significantly diminished increases in cellular ROS, LPOs, and LIP, while restoring GSH content and GPX–4 activity. Additionally, it suppressed the cytokine production of 5–FU-induced cells in a concentration–dependent manner. Moreover, TS exerted wound-healing effects against skin injuries and 5–FU damage significantly and dose dependently. Conclusions: The insights of this work have identified biochemical mechanisms using tea saponin extract to protect against 5–FU-induced keratinocyte ferroptosis and inflammation. This study highlights the promising adjunctive potential of tea saponin in the mitigation and management of chemotherapy-induced mucositis.

## 1. Introduction

Cancer treatment, while advancing rapidly, can often bring about severe side effects that compromise the quality of life of patients. More than 75% of those receiving chemotherapy will experience chemotherapy-induced oral mucositis as a significant clinical issue that manifests as painful inflammation and ulcerations of the mucous membrane, specifically in the gastrointestinal tract. This condition is predominantly caused by the administration of chemotherapeutic agents like 5–fluorouracil (5–FU), a widely used anticancer drug [1,2,3,4]. The underlying mechanisms of 5–FU that contribute to mucositis include cell death, ferroptosis, and lipid–peroxidation (LPO)-driven cell death, which is increasingly recognized as a crucial factor in the pathogenesis of this condition [5,6]. Despite progressive understandings of the mechanisms of 5–FU-induced mucositis, there is a pressing need for effective therapeutic strategies to mitigate its effects. Current treatments have primarily focused on symptomatic relief, leaving the root cause largely unaddressed [7,8].

Skin inflammation can be caused by ultraviolet B radiation, diabetes, psoriasis, 5–FU, and erastin, leading to ferroptosis and many cutaneous diseases. Ferroptosis is a distinctive mode of cell death that holds promise for developing innovative therapeutic strategies. It can target many diseases, including sepsis, mucositis, inflammatory disorders, chronic diseases, and cancer. In this context, exploring compounds with the potential to inhibit ferroptosis could offer a novel approach to managing mucositis.

Saponins are amphiphilic glycosides of triterpenes and steroids that exhibit soap-like properties, in which the molecule has highly polar sugar moieties together with non-polar triterpene or sterol backbones. In ferroptosis, saponin may interact with certain signaling pathways, including the cystine/glutamate antiporter, as system Xc–glutathione (GSH)–glutathione peroxidase 4 (GPX–4), nuclear factor erythroid 2-related factor 2 (Nrf2), tumor protein p53, and mammalian target of rapamycin (mTOR) [9]. Interestingly, Anemoside B4 is a triterpenoid saponin that can inhibit an inflammasome-mediated inflammatory response, increase antioxidant capacity, and decrease mitochondrial reactive oxygen species (ROS). It can also decrease the level of glycogen synthase kinase 3beta and ferroptosis [10]. In addition, Astragaloside, which is a triterpenoid saponin found in the root of the Astragalus membrane, has exhibited anti-inflammatory, antioxidative, and certain protective effects against cisplatin-induced liver injuries [11]. Moreover, theasaponins E_1_ and E_2_ are found in the seeds; theasapogenols, sapogenins and camelliagenins in the leaves; olean–12–ene type triterpenoid saponins in the roots; and camsinsaponins and camellioside in the flowers of *Camellia species* tea [12,13,14,15]. Regarding their benefits, tea saponins (TS) can serve as natural surfactants, antioxidants [16], stabilizers for the production of hesperidin nanoparticles [17], as well as anti-microbial, anti-allergic, anti-inflammatory and anti-photoaging agents [14,15,18,19]. However, the role of tea saponin against 5–FU-induced ferroptosis in keratinocytes has not yet been thoroughly investigated. This study aimed to examine the protective effects of tea saponin on 5–FU--induced ferroptosis and any related damage to human keratinocyte (HaCaT) cells.

## 2. Materials and Methods

### 2.1. Chemicals and Reagents

Bovine serum albumin (BSA), L–ascorbic acid, dimethyl sulfoxide (DMSO), erastin, ferrous ammonium sulfate (FAS), fetal bovine serum (FBS), 3–(4,5–dimethylthiazol–2–yl) 2,5–diphenyl tetrazolium bromide (MTT), and α–tocopherol were purchased from Sigma-Aldrich Company (Limited), St. Louis, MO, USA. Dulbecco’s Modified Eagle’s Medium (DMEM), Hanks’ balanced salt (HBS), 4–(2–hydroxyethyl)–1–piperazineethanesulfonic acid (HEPES), along with pH 7.0 buffer, radioimmunoprecipitation assay (RIPA) lysis buffer, 0.25% trypsin/0.02% disodium ethylenediamine tetraacetate (trypsin–EDTA), Hoechst nuclear dye 33342, and phosphate-buffered saline pH 7.4 (PBS) solution were obtained from Thermo–Fisher Scientific Company, Waltham, MA, USA. Furthermore, 2′,7′–Dichlorodihydrofluorescein diacetate (H_2_DCFDA) was obtained from Invitrogen Molecular Probes Company, Eugene, OR, USA. Moreover, *N*–(4–Diphenylphosphinophenyl)–*N′*–(3,6,9,12–tetraoxatridecyl)perylene–3,4,9,10–tetracarboxydiimide (Liperfluo), and Mito-FerroGreen fluorescence probes were purchased from Dojindo Laboratory Company (Limited), Kamimachiki-gun, Kumamoto, Japan. Cytosense LI, Mitosense L1, and 3–[4–(perylenylphenylphosphino)phenoxy]propyltriphenylphosphonium iodide (MitoPeDPP) fluorescence probes were purchased from MINT Therapeutics (London, UK, https://www.mint-therapeutics.com). An annexin V–AbFluor™ 488/PI apoptosis detection kit (Product number: KTA0002) was purchased from Abbkine Scientific Company (Limited), Atlanta, GA, USA. A glutathione (GSH) assay kit (Product code: E-BC-K030-S) and a glutathione peroxidase 4 (GPX–4) activity assay kit (Product code: E–BC–K883–M) were purchased from Elabscience Biotechnology Company (Limited), Wuhan, China. Quick Start™ A Bradford protein assay kit was purchased from Bio–Rad Laboratories Company (Limited), Hercules, CA, USA. Sandwich enzyme-linked immunoassay (ELISA) kits for interleukin 6 (IL–6) (Product code: SEKB10395) and tumor–necrotic factor alpha (TNF-α) (Product code: SEKA10602) were obtained from Shino Biological Incorporation, Kechuang, Beijing, China. Penicillin, streptomycin, 5–FU, and desferrioxamine mesylate (DFO, Novartis, Basel, Switzerland) were obtained from a local drugstore, Maharaj Nakorn Chiang Mai Hospital, Faculty of Medicine, Chiang Mai University, Chiang Mai, Thailand.

### 2.2. Crude Tea Saponin

Tea saponin extract (TS) was purchased from Xi’an International Healthcare Factory Company (Limited), Xi’an, China. The product was prepared from *Camellia oleifera* tea seeds according to the manufacturer’s protocol. The resulting product appeared as a dark-yellow powder (80-mesh particle size) that had a bitter flavor and taste. According to HPLC analysis, 100 g of food-grade TS contained 83.9 g of total saponin, 32.5 g of camelliasaponins A, 30.5 g of camelliasaponins B, 9.5% of metesaponin, and 6.5 g of assamsaponin, as well as 4.9 g of phenolic and flavonoid compounds. The shelf-life can be up to 24 months when stored at 4 °C in a refrigerator.

### 2.3. Keratinocyte Culture

Immortalized human keratinocytes (HaCaT cells) were obtained from a nonprofit American Type Culture Collection (ATCC) Organization, in Manassas, VA, USA. They were maintained in DMEM supplemented with 10% (*v*/*v*) FBS and 1% (*w*/*v*) penicillin/streptomycin called complete medium at 37 °C in a humidified atmosphere with a 5% CO_2_ incubator. HaCaT cell suspension (100 μL) was seeded in each well of a 96-well plate (a density of 1 × 10^4^ cells/well) and incubated at 37 °C in a 5% CO_2_ incubator for 24 h. Upon 80% confluence, the cells were washed once with PBS solution and prepared in a suspension with the culture medium for further study.

### 2.4. Cytotoxicity Test

Cell viability was determined using MTT dye, which was based on the mitochondrial reductase enzymes of viable cells to reduce the yellow MTT substrate to a purple formazan product [20]. In the assay, HaCat cells were treated with PBS and TS (6.25–100 µg/mL) solution with/without 5–FU (5 µg/mL) at 37 °C for 24 h. Afterward, MTT reagent was added to each well of the treated cells, and they were incubated at 37 °C in a 5% CO_2_ incubator for 4 h. Finally, the produced formazan crystals were solubilized with 0.1% DMSO (100 μL/well) and optical density (OD) was measured at a wavelength of 570 nm against the reagent blank using a microplate reader (Synergy H4, BioTek Instruments, Winooski, VT, USA). The OD values correlated with the number of viable cells, allowing for the quantitative analysis of cytotoxicity.

### 2.5. Measurement of Cellular ROS

Intracellular ROS values were determined using an oxidation-sensitive H_2_DCFDA fluorescent probe [21,22]. HaCat cells were seeded onto a 96-well plate and treated with TS (3.13, 6.25. 12.5, and 18 µg/mL) or L–ascorbic acid (50 µg/mL) in the presence of 5–FU (5 μg/mL) in a 5% CO_2_ incubator at 37 °C for 24 h. They were sedimented using a centrifuge (Type EBA200, fixed-angle rotor, Andreas Hettich GmbH Company, Tuttlingen, Germany) at 4 °C and washed twice with PBS. Afterward, 20 μM H_2_DCFDA solution was added to stained cytosolic ROS and 10 mM diamino propidium iodide to stain the nucleus of the treated cells at 37 °C for 30 min. Finally, green fluorescence of oxidized dichlorofluorescein (λ_excitation_/λ_emission_ values of 488 nm/530 nm) and intense blue fluorescence of Hoechst dye 33,342 (λ_excitation_/λ_emission_ wavelengths of 350 nm/465 nm) imaging (5 µm scale bars) were administered using a scanning confocal fluorescence microscope (Zeiss Model LSM 710, Carl Zeiss AG Meditec, Oberkochen, Germany). Quantitative fluorescence intensity (FI) was measured using a 96-well plate spectrofluorometer (Synergy H4, BioTek Instruments, Winooski, VT, USA) at λ_excitation/_λ_emission_ values of 485 nm/525 nm.

### 2.6. Assessment of Cellular LPOs Content

#### 2.6.1. Membrane LPOs

Membrane LPOs were detected using a Liperfluo fluorescence probe [23]. HaCat cells were treated with TS (3.13, 6.25.12.5, and 18 µg/mL) or α–tocopherol (50 µg/mL) in the presence of 5–FU (5 μg/mL) at 37 °C in a 5% CO_2_ incubator for 24 h. The treated cells were then stained with 1 μM Liperfluo solution previously dissolved in 0.1% DMSO for 30 min at 37 °C in the dark and washed twice with HBS solution. Afterward, the cells were detached from the wells using sterile-filtered trypsin/EDTA solution, harvested into 1.5 mL microtubes, and sedimented gently using a centrifuge (Type EBA200, fixed-angle rotor, Andreas Hettich GmbH Company, Tuttlingen, Germany) at 300× *g* at 37 °C for 10 min. After removing the supernatant, the cell pellet was gently resuspended in a pre-warmed HBS solution, and the FI value of 10,000 cells was measured with a flow cytometer (Beckman Coulter Life Sciences, Indianapolis, IN, USA) at λ_excitation_/λ_emission_ values of 488 nm/550 nm. Data were collected and analyzed using CytExpert for DxFLEX 2.0 software (Beckman Coulter Life Sciences, Indianapolis, IN, USA).

#### 2.6.2. Mitochondrial LPOs

MitoPeDPP is a cell membrane-permeable probe that specifically localizes in the mitochondria due to the triphenylphosphonium moiety and detects mitochondrial LPOs [24]. The treated cells were stained with a MitoPeDPP fluorescence probe (1 mM) and FI levels were then measured (10,000 cells/sample) at λ_excitation_/λ_emission_ values of 452 nm/470 nm using a Beckman Coulter flow cytometer. Data were collected and analyzed using CytExpert for DxFLEX 2.0 software, as has been described above.

### 2.7. Detection of Cellular LIP

#### 2.7.1. Cytosolic LIP

Cytosense LI is a highly fluorescent iron sensor containing bifunctional iron chelator and fluorophore structures that can enter cells and selectively bind redox-active LIP, consequently quenching the FI [25]. Cytosense LI solution (70 µM) was freshly prepared in 0.1% DMSO according to the manufacturer’s protocol. In the baseline assessment, HaCat cells (1.8–2 × 10^5^ cells/well) were incubated at 37 °C in a 5% CO_2_ incubator overnight with 100 µM DFO to deplete cellular LIP or PBS (control) and then stained with 70 µM Cytosense LI solution at 37 °C for 2 h to enable detection of the cytosolic iron. The cells were then treated with 5–FU (5 μg/mL) and TS (3.13 and 6.25 µg/mL) solution at 37 °C in a 5% CO_2_ incubator for 4–5 h and stained with 70 µM Cytosense LI solution at 37 °C for 2 h. Afterward, the treated cells were detached from the wells with trypsin/EDTA solution, resuspended in 10 mM HEPES pH 7.0 containing 150 mM NaCl, transferred to a 96-well black plate, and FI levels were then measured at λ_excitation_/λ_emission_ values of 430/473 nm using the BioTek spectrofluorometer. The FI differences (∆FI) between the DFO-treated and untreated cells were utilized to quantify cytosolic LIP, using an ex situ calibration curve prepared with different FAS concentrations.

#### 2.7.2. Mitochondrial LIP

A Mito–FerroGreen fluorescence probe was used to quantify mitochondrial Fe^2+^ [26]. According to the manufacturer’s protocol, 1 mM Mito–FerroGreen stock solution was firstly prepared in 0.1% DMSO and then diluted with HBS solution to prepare 5 μM Mito–FerroGreen working solution. HaCat cells were treated with PBS (control), erastin (20 µM), α–tocopherol (50 µg/mL), 5–FU (5 μg/mL), and TS (3.13–12.5 µg/mL) at 37 °C in a 5% CO_2_ incubator for 24 h. The treated cells were incubated with 5 μM Mito–FerroGreen solution at 37 °C in a 5% CO_2_ incubator for 30 min and fluorescence imaging was administered at λ_excitation_/λ_emission_ values of 488 nm/530 nm using a Zeiss confocal fluorescence microscope (Carl Zeiss Microscopy GmbH, Jena, Germany).

### 2.8. Determination of GSH Content

The amount of GSH was determined using the colorimetric assay kit following the manufacturer’s instructions. HaCaT cells (1 × 10^6^ cells/well) were treated with PBS, erastin (20 µM), 5–FU (5 μg/mL), and TS (3.13–12.5 µg/mL) at 37 °C in a 5% CO_2_ incubator for 24 h. The treated cells were harvested and subjected to RIPA lysis buffer (0.5% deoxycholate, 1% Nonidet P-40, 0.1% sodium dodecyl sulfate, 100 μg/mL of phenylmethylsulfonyl fluoride, 1 mM Na_2_VO_4_, and 8.5 μg/mL of aprotinin in PBS). The treated cells were then shaken for 20 min at 4 °C. The cell lysates were centrifuged to remove any debris, and the supernatants were collected for analysis of GSH content and GPX–4 activity according to the manufacturer’s instructions.

The glutathione assay reaction was initiated by mixing the cell lysates or GSH standards with the substrate solution in a 96-well plate, followed by incubation at room temperature for an indicated period. OD values were measured at 405 nm against a reagent blank using a BioTek 96-well microplate reader (BioTek Instruments, Winooski, VT, USA). A calibration curve of GSH was made from different standard GSH concentrations (5–200 mg/mL). Accordingly, GSH contents in the lysates were determined from the calibration curve and were normalized for protein content to be measured below.

### 2.9. Assay of GPX–4 Activity

A GPX–4 assay kit was used as per the manufacturer’s protocol, in which GPX–4 catalyzed the conversion of H_2_O_2_ and GSH substrates to H_2_O and oxidized glutathione (GSSG) products. The resulting products then consumed the reducing agent NADPH_2_ (λ_maximum_ at 340 nm) with the addition of glutathione reductase (GR). The GPX–4-specific activity was calculated by measuring nonspecific GPX–4 activity and total GPX–4 activity by adding a GPX–4 inhibitor to the system. In our analysis, the cell lysate was diluted at a ratio of 1:1 (*v*/*v*) with 0.9% NaCl along with a stabilizer solution and centrifuged (Type EBA200, fixed-angle rotor, Andreas Hettich GmbH Company, Tuttlingen, Germany). The clear supernatant was added to the sample well and the control well (20 µL each), which was followed by the addition of the reaction solution into the sample well and the inhibitor solution into the control well (40 μL each). They were mixed well for 5 s and incubated at room temperature for 15 min. Finally, the working GR solution (40 μL each) was added to all wells. OD values of NADPH were measured immediately at 340 nm with a BioTek microplate reader and recorded. They were found to directly correlate with GPX–4 activity. Accordingly, the specific activity of GPX–4 (units/mg of protein) was calculated using the designated formula (1):Specific activity = (∆OD_sample_ − ∆OD_control_) ÷ (ε × d) × (V_total_ ÷ V_sample_) ÷ T × 2 ÷ C_protein_(1)
where ∆OD_sample_ is the change of the OD value of the sample well; ∆OD_control_ is the change of the OD value of the control well; ε is the molar extinction coefficient of the product at 340 nm, 6.22 × 10^−3^ L/μmol/cm; d is the optical path of cuvette, 0.6 cm; V_total_ is the total volume of reaction, 0.24 mL; V_sample_ is the volume of sample, 0.02 mL; T is the time of lysate sample reaction, 15 min; C_protein_ is the protein concentration in the lysate sample, g/L; and 2 is the dilution factor of the sample before testing. In addition, the quality control of the assay kit indicated a sensitivity value of 3.22 U/L, a detection range of 3.22–44.69 U/L, an inter-assay coefficient of variation value in the range of 0.4–3.4%, and an intra-assay CV value in the range of 1.2–3.1%.

### 2.10. Measurement of Protein Content

Lysate protein was determined using Bradford’s Coomassie brilliant blue dye-binding method [27] according to the manufacturer’s protocol. In the assay, 100 µL of cell lysate and BSA standard were incubated with 1.0 mL of Bradford reagent at room temperature for 5 min, and the OD values of the product were measured at 595 nm against a reagent blank. The amount of protein was then determined from a calibration curve made from the BSA solution (0.125–2 mg/mL).

### 2.11. Assessment of Apoptotic Cells

Apoptotic keratinocytes were assayed using the Annexin V–AbFluor™ 488/DAPI apoptosis detection kit (Abcam Ltd., Cambridge, UK) [28] following the manufacturer’s protocol. HaCaT cells (2 ×10⁵ cells/well) were treated with erastin (10 µM) alone and 5–FU (5 µg/mL) in the presence of TS (0–12.5 µg/mL) at 37 °C in a 5% CO_2_ incubator for 16–24 h and then harvested with trypsinization. After being washed twice with PBS, the treated cells were resuspended in Annexin V binding buffer and stained with Annexin V–AbFluor™ 488 (5 µL) and DAPI (2 µL) reagents per 100 µL cell suspension in the dark for 15 min at room temperature. Finally, the F1 values of the stained cells were analyzed within 30 min using flow cytometry at λ_excitation_/λ_emission_ wavelengths of 491 nm/517 nm for AbFluor™ 488 and 535 nm/617 nm for DAPI.

### 2.12. Quantification of Inflammatory Cytokines

Human IL–6 and TNF–α concentrations were measured using sandwich ELISA Pair Sets according to the manufacturer’s instructions. Firstly, the 96-well microplates were coated with mouse capture monoclonal antibodies that were specific to human IL–6 (2 µg/well) and TNF–α (2 µg/well), having been previously dissolved in PBS pH 7.4 overnight at 4 °C. They were then washed twice with PBS containing 0.05% Tween–20, and the non-specific binding sites were blocked with 2% (*w*/*v*) BSA solution. Similarly, HaCat cells were treated with 10 µM erastin or and 5–FU (5 µg/mL) in the presence of TS (3.13, 6.25, and 12.5 µg/mL) at 37 °C in a 5% CO_2_ incubator for 16–24 h. After incubation, the treated cells were sedimented with the centrifuge, and the culture medium was collected and diluted at a ratio of 1:20 with the dilution solution for quantitation of secretory IL–6 and TNF–α. In analysis, diluted samples, standard IL–6, and standard TNF–α (100 µL each) were added to the two antibodies-precoated wells, and they were incubated at room temperature for 2 h. After being washed with PBS containing 0.05% Tween–20 solution, horseradish peroxidase (HRP)-conjugated detection antibodies specific for human IL–6 (0.25 µg/mL) and TNF–α (0.5 µg/mL) were added to all wells. They were then shaken gently and incubated at room temperature for 1 h. After being washed twice, tetramethylbenzidine containing 3% H_2_O_2_ substrate solution (200 µL) was added to all wells, and the reaction was stopped after 20 min using 2 N H_2_SO_4_. Finally, OD values were measured at 450 nm using the BioTek 96-well microplate reader (BioTek Instruments, Winooski, VT, USA). Standard curves of IL–6 (3.13–200 pg/mL) and TNF–α (23.44–1500 pg/mL) were constructed and used to determine their concentrations in the samples.

### 2.13. Scratch Assay

Different concentrations of TS were prepared in a sterile PBS buffer. HaCaT cells were cultured in a sterile Falcon 6-well plate (Corning Technologies, Glendale, AZ, USA) and seeded at a density of 4 × 10^4^ cells/well with 85–90% confluence. A scratch was made across the cell monolayer using a sterile pipette tip, and the scratched cell layer was rinsed with sterile PBS to remove the detached cells. Next, the scratched cells were induced with 5–FU (5 μg/mL) and treated with TS (3.13, 6.25, 12.5 and 25 µg/mL) or L–ascorbic acid (50 µg/mL) (1 mL each) at 37 °C for 24 h. Finally, cell images were taken with an inverted light microscope (Olympus ΙX-81, Olympus Optical Corporation Company, Hachioji, Tokyo, Japan) equipped with a digital camera (Canon EOS 700D, Canon Incorporation, Ota, Tokyo, Japan) at 0, 24, 48, and 96 h after the compound treatments had been administered. Cell image acquisition was carried out with Analysis get IT software version 1.2.2 (Olympus Software Imaging Solutions, Münster, Nordrhein-Westfalen, Germany). The scratch area was measured and analyzed using free, open-sourced Image J2 processing software version 2.16.0 supported by the National Institutes of Health [29,30].

### 2.14. Statistical Analysis

Descriptive analysis of results and graph plotting were performed using the GraphPad Prism 10 Software (GraphPad Company, Boston, MA, USA) for Windows (IBM Corporation, Armonk, NY, USA). Quantitative variables have been expressed as mean ± standard deviation (SD). One-way analysis of variance (ANOVA), followed by post-hoc Duncan tests for parametric data, were used to compare the two groups to determine the statistical significance. A *p*-value < 0.05 was considered statistically significant.

## 3. Results

### 3.1. Protective Effects of TS Against 5–FU-Induced Cytotoxicity

As is shown in Figure 1, TS doses up to 25 µg/mL were not harmful to HaCaT cells, whereas 50 and 100 µg/mL TS dramatically declined the cell viability to 4%, of which a half-maximal inhibitory concentration (IC_50_) of TS was found to be 43.6 µg/mL, indicating moderate-dose cytotoxicity when used alone. Induction of 10 µg/mL 5–FU was toxic to HaCaT cells and decreased cell viability to 75%, showing an IC_50_ value of 31 µg/mL; nevertheless, TS treatments were unable to protect the toxicity of HaCaT cells induced by 5–FU. The findings suggest that TS concentrations < 25 µg/mL should be safe to treat epithelial HaCaT cells without/with 5 µg/mL 5–FU induction.

### 3.2. Reduction of 5–FU-Induced Oxidative Stress by TS

As can be seen in green DCF fluorescence imaging (Figure 2A), treatment of HaCaT cells with 5 µg/mL 5–FU significantly increased ROS levels when compared with control cells, while TS (3.13–12.5 µg/mL) intervention restored the increased ROS levels in a concentration-dependent manner. Concomitantly by blue Hoechst fluorescence imaging, 5–FU treatment caused nuclear DNA damage, while the TS intervention diminished the increased DNA damage (less blue fluorescence). Regarding quantitative FI, TS intervention, as well as L–ascorbic acid (50 µg/mL), dose dependently reduced cellular ROS levels when compared with samples that had not received treatment (*p* < 0.005), for which the 12.5 µg/mL TS was found to be the most effective (Figure 2B,C). Thus, TS treatment promoted antioxidant defenses and scavenged the generated ROS, while protecting against oxidative damage of epithelial HaCaT cells under 5–FU chemotherapy.

### 3.3. Inhibition of Lipid Peroxidation by TS

Levels of LPOs are shown in terms of FI peak (Figure 3A) and quantitative lipid peroxidation (Figure 3B). An induction of 5 µg/mL 5–FU dramatically increased LPOs levels in HaCaT cells when compared with control cells (*p* < 0.005). Considerably, TS treatment as well as 100 µg/mL α–tocopherol restored the increased cellular LPOs levels in a dose-dependent manner (*p* < 0.005) when compared with samples that were not treated. The results implied that TS could effectively present anti-lipid peroxidation activity to protect against oxidative membrane damage and suppressed ferroptosis of HaCaT cells induced by 5–FU.

In addition, MitoPeDD green-fluorescence cell imaging implied that mitochondrial lipid peroxidation, ferroptosis, and Hoechst blue-fluorescence cell imaging can indicate DNA damage (Figure 4A). Regarding the quantitative results (Figure 4B), green FI was the brightest and most substantially increased in HaCaT cells treated with 5 µg/mL 5–FU, and 50 µM erastin (*p* < 0.005) when compared with control cells. Importantly, TS (3.13–12.5 g/mL) treatment reinstated increased LPO levels depending upon the concentrations, whereas α–tocopherol (100 µM) treatment restored the increased LPOs to baseline levels (*p* < 0.005). The results indicated that TS and α–tocopherol exhibited the inhibitory effect of anti-lipid peroxidation against ferroptosis in HaCaT cells induced by 5–FU and erastin.

### 3.4. Effect of TS Treatment on Cellular LIP

The amount of LIP in HaCaT cells induced by 5-FU (5 µg/mL) was significantly increased when compared with control cells and when the increased LIP was reinstated significantly by TS (3.13 and 6.25 µg/mL) treatment, for which 3.13 µg/mL TS was found to be more effective than 6.26 µg/mL TS (Figure 5A). As illustrated in Figure 5B, the amount of LIP was decreased in 100 µM DFO-chelated cells (bar 2) but increased in 5–FU (5 µg/mL)-induced cells (bar 3) when compared with the control cells (bar 1). In addition, DFO (100 µM) chelation (bar 4) and TS (3.13 and 6.25 µg/mL) treatments (bars 5 and 7) diminished the amounts of LIP in HaCaT cells induced by 5–FU (5 µg/mL) alone. Nevertheless, combined treatments of DFO with the two TS doses no longer changed the LIP levels (bars 6 and 8) when compared with their single therapies. The results suggest that TS treatments like DFO attenuate cellular iron homeostasis and alleviate iron-catalyzed oxidative damage, thereby preventing ferroptosis in 5–FU-induced HaCaT cells.

Similarly, treatments with 50 µM erastin and 5 µg/mL 5–FU decreased mitochondrial LIP levels significantly, whereas TS treatment significantly restored the increased LIP independently of the concentrations (Figure 6A–C).

### 3.5. Restoration of Cellular GSH and GPX–4 Levels by TS

In our findings, both GSH content and GPX–4 specific activity were depleted in 5 µg/mL 5–FU and 50 µM erastin-treated HaCaT cells significantly when compared with the control cells. TS (3.13–12.5 µg/mL) treatment significantly repleted the decreased GSH and GPX–4 values in a concentration-dependent manner (Figure 7A,B). Notably, the most substantial recovery was observed at 6.25 µg/mL. Accordingly, TS played an important role in enhancing cellular antioxidant defense by recycling GSSG to GSH and improving GPX–4 activity to overcome excessive ROS production catalyzed by 5–FU chemotherapy, while also counteracting any ferroptosis inducer such as erastin.

### 3.6. Anti-Ferroptosis Effects of TS Against Erastin- or 5–FU-Induced HaCaT Cells

As has been shown above, 5 µg/mL 5–FU treatment of HaCaT cells caused cell ferroptosis, which was characterized by increases in cell death, ROS production, LPO accumulation, redox-active iron, and depletions of GSH and GLX–4 activity. Flow cytometric results revealed that ferroptosis cell populations were increased to 35.12% by 10 µM erastin and 30.88% by 5 µg/mL 5–FU induction when compared with 12.64% for the control cells, and the cell deaths were decreased to 18.94%, 13.40%, and 10.92% by TS intervention at 3.13, 6.25, and 12.5 µg/mL, respectively (Figure 8A). In our quantitative results, HaCaT cell death was increased by erastin and 5–FU treatments (*p* < 0.005), while the ferroptosis cell deaths were dramatically decreased by TS intervention in a concentration-dependent manner (*p* < 0.005) (Figure 8B). The findings are suggestive of the potential effect of TS against ferroptosis cell death in 5–FU-induced HaCaT cells and its promising therapeutic applications for mitigating chemotherapy-associated epithelial damage.

### 3.7. Anti-Inflammatory Activity of TS

The potential anti-inflammatory property of TS was evaluated by measuring levels of pro-inflammatory cytokines. It was found that the application of 5 µg/mL 5–FU to HaCaT cells significantly increased the secretion and production of IL–6 and TNF–α, reflecting the inflammatory response induced by the chemotherapeutic drug. TS (3.13–18 µg/mL) and 50 µg/mL L–ascorbic acid significantly reduced the increased IL–6 and TNF–α concentrations in a concentration-dependent manner (Figure 9A and Figure 9B, respectively). Accordingly, TS exerted a potent anti-inflammatory activity against 5–FU-induced epithelial cell damage.

### 3.8. Wound-Healing Effects of TS

The wound-healing capacity of TS was evaluated using a scratch assay. The results revealed that 5–FU deteriorated wound closure of HaCaT cells, whereas TS significantly promoted cell migration and wound closure in a concentration- and time-dependent manner when compared with samples that were not treated (Figure 10). The results suggest the promising therapeutic application of TS in alleviating chemotherapeutic-induced epithelial injuries and mucositis.

## 4. Discussion

Keratinocytes and macrophages are involved in the pathogenesis chronic skin inflammation through the production of different chemokines and cytokines in response to stimulation by pro-inflammatory mediators such as TNF–α/interferon–gamma (IFN–γ) and lipopolysaccharide (LPS). Extrinsic factors, such as ultraviolet A and B (UVA and UVB) irradiation, 2,4–dinitrochlorobenzene (DNCB) topical applications, and chemotherapeutic 5–FU treatment, can cause skin keratinocyte damage and pathogenesis. In oncology, 5–FU is a halogenated pyrimidine nucleoside analog that is currently being used in clinics as a form of anticancer chemotherapy because it can inhibit thymidylate synthase in DNA/RNA synthesis in tumor cells. However, certain side effects, such as oral mucositis, have been reported [31].

Saponins are amphiphilic glycosides composed of aglyconic triterpenoids or steroids linked to oligosaccharide moieties and secondary metabolites that are naturally found in the leaves, flowers, and fruits of tea, ginseng, and other plants. Importantly, they serve as an element of the plant’s defense systems against diseases and herbivores, while also being recognized as allelopathic agents between plants and pharmaceuticals [32]. In *Saponaria species* varieties, *Saponaria officinalis* root extract was found to contain six major saponins (e.g., gypsogenin and gypsogenic acid derivatives, as well as saponariosides C, D, and E) and six phenolic compounds (e.g., rutin, quercetin galactoside, syringic acid, apigenin, protocatechuic, and vanillic acid) that have been found to exert antioxidant, antibacterial, and anticancer activities [33]. Zhu and colleagues have previously reported that the methanolic extraction of *C. oleifera* seeds revealed 25.24% yield and 36.15% purity values, while the ammonium sulfate/propanol extraction revealed a purity value of 83.72% [34]. In our analysis, *C. oleifera* seed and root extracts contained sasanquasaponin and oleiferasaponins (e.g., oleanane and oleiferosides) [34,35]. These ingredients increased the extract’s viability in a concentration-dependent manner when used at up to 10 µM. They have also been found to reduce malondialdehyde content; improve GSH content; and enhance the activities of GPX, superoxide dismutase (SOD), and catalase (CAT) in rat cardiomyocytes [36,37]. Inevitably, the tea seed extract elucidated its toxic effects against human hepatocarcinoma BEL–7402, gastric carcinoma BGC–823, breast cancer MCF–7, promyelocytic leukemia HL–60, and oral epithelial KB cell lines with IC_50_ values of 11.02, 5.98, 10.61, 2.55, and 19.76 µg saponins/mL, respectively [37].

In the present study, crude saponin extract derived from *C. oleifera* seeds was comprised of saponins, phenolics, and flavonoids. It was investigated for its cytotoxicity and free-radical scavenging activity, as well as its anti-ferroptosis, anti-apoptosis, anti-inflammatory, and wound-healing effects on 5–FU-induced human keratinocyte HaCaT cells. We found that TS treatment at doses of 6.25–25 µg/mL was not toxic to HaCaT cells and lowered cell survival when used at doses of 50 and 100 µg/mL (IC_50_ = 43.6 µg/mL). Moreover, 5 µg/mL 5–FU exerted potent toxicity toward HaCaT cells, resulting in 75% cell survival; nonetheless, TS did not protect cells from 5–FU-induced cellular injury. Consistently, crude methanolic extract of the *Verbascum nigrum* aerial part, and its six fractions, used at concentrations in a range of 25–150 µg/mL did not influence the viability of HaCaT cells [38]. In contrast, methanol extracts of the *Glycyrrhiza-glabra* roots, that were comprised of triterpene saponin glycyrrhizin, flavonoids, and isoflavonoids, were found to be toxic to HaCat cells, indicating a viability of 35% when used at 250 µg/mL. However, the IC_50_ value was 242 µg/mL when administered at doses in a range of 15.6–250 µg/mL [39].

Under oxidative stress, excess ROS may destroy certain cellular biomolecules, such as carbohydrates, proteins, lipids, and nucleic acids, which can lead to metabolic syndromes, cardiovascular diseases, cancers, aging, and many other conditions. Accordingly, antioxidant compounds, such as L–ascorbic acid, α–tocopherol, N–acetylcysteine, phenolics, flavonoids, and other phytochemicals, are required to scavenge overwhelming free radicals. Interestingly, triterpenoid saponin extracts derived from *C. japonica* roots have been reported to induce nuclear factor erythroid 2-related factor 2 activity against oxidative damage in HaCaT cells by upregulating the expression of antioxidant response element genes in the gene promoters [40].

Herein, we revealed that *C. oleifera* saponin product exerted anti-ferroptosis activity against 5–FU-induced HaCaT cells by decreasing levels of cytosolic and mitochondrial ROS, redox-active iron, and LPOs while increasing levels of cellular GSH content and GPX–4 activity. In comparison, *C. japonica* camellioside, which is a triterpene saponin, could restore increased matrix metalloproteinase–1 production, the suppression of type I alpha1 pro-collagen, macrophage-associated protein kinase (MAPK) activation, and the consequential phosphorylation of c–Fos and c–Jun in UVA-irradiated HaCaT cells [15]. Regarding the antioxidant and inflammasome inhibitors, saponin and maltol obtained from Korean red ginseng extracts could mitigate mitochondrial oxidative stress and inhibit IL–1β secretion of HaCaT cells by interfering with the upregulation of the pro-IL–1β and nucleotide-binding domain while contributing to the leucine-rich-containing family and the pyrin domain-containing-3 genes mediated by a toll-like receptor 3 ligand [41,42]. In addition, ginseng berry saponins could prevent UVB-induced injuries in human skin keratinocytes by improving HaCaT viability, thereby decreasing DNA damage, mitochondrial oxidative stress, and LPO production. Additionally, they restored the activities of certain antioxidant enzymes, including GPX, superoxide dismutase, and catalase, which had diminished the erastin- and iron-induced ferroptosis of HaCaT cells [43]. Proposedly, TS could effectively exhibit free-radical scavenging activity since the compound contains many hydroxyl groups as electron donor(s). Considering their log *p* values, 5–FU and erastin may be easily incorporated into the cells and increase ROS generation and the dysfunction of mitochondria (inhibition of manganese superoxide dismutase and reductase system). The two chemicals could also induce certain forms of oxidative stress such as GSH depletion and GPX–4 inhibition. The present work has revealed that the levels of apoptotic markers, such as phosphatidylserine exposure on HaCaT cells, were increased by erastin and 5–FU treatments and were dose-dependently declined by TS interventions. Previous findings have indicated that erastin and 5–FU induced apoptotic cell death via the mitochondrial intrinsic pathway in keratinocytes, suggesting that the type of cell death induced may depend upon the cell types and the chemical structure of the inducers [44,45].

Ginsenoside Rg5:Rk1 derived from *Panax ginseng* exhibits potential anti-AD properties by suppressing the nuclear factor kappa B (NF–κB)/p38 MAPK/signal transducer and by serving as an activator of transcription 1 signaling in TNF–α/IFN–γ-stimulated HaCaT cells and LPS-stimulated macrophages [46]. In addition, saikosaponins, which are bioactive oleanane-type triterpenoids found in *Radix bupleuri* plants, could express anti-inflammatory effects by downregulating the expression of the early growth response 1 (EGR1) transcription factor. This was done via inhibition of the extracellular signal-regulated kinase (ERK) 1/2, c–Jun N–terminal kinase (JNK) 1/2, and the p38/MAPK pathways in HaCaT cells, which then helped to lower skin lesions and recover impaired filaggrin protein levels in the inflamed skin of 2,4–dinitrochlorobenzene-induced BALB/c mice [47,48]. Moreover, treatment with *C. oleifera* sasanquasaponin extract (30 µg/mL) decreased the production of ROS, IL–1β, IL–6, and TNF–α in LPS-induced RAW264.7 cells, which indicated its free radical-scavenging and anti-inflammatory activity.

Inversely, platycodin D, a triterpene saponin, is a major constituent in the roots of the *Platycodon grandiflorum* medicinal plant that could induce apoptosis in HaCaT cells via DNA fragmentation, caspases–3 and 8 activations, activation of the inhibitor of NF–κB kinase–beta, and upregulation of the Fas receptor and Fas ligand expression [49]. Remarkably, ethanol extracts of *C. oleifera* tea buds containing flavonoids, phenols, and terpenoids were toxic to non-small lung cancer A549 cells (IC_50_ = 57.53 µg/mL) and induced cell apoptosis by diminishing membrane potential, upregulating Bax, and activating caspase 9 and caspase 3 activity of the mitochondria [50].

Regarding their wound-healing effects, we have demonstrated that tea saponins could promote cell migration and wound closure in 5–FU-induced HaCaT cells. Similarly, saponin, a ginsenoside Rb1 extract derived from red ginseng roots, could enhance vascular endothelial growth factor (VEGF) production and hypoxia-inducible factor 1alpha 4 expressions in IL–1β-induced HaCaT cells, which could also increase neovascularization and the production of VEGF and IL–1β in the burn wounds of mice [51]. In contrast, a triterpenoid saponin, astragaloside, was found to increase EGFR and ERK activities, cell proliferation, and migration in HaCaT cells, while also contributing to the healing of wounds in mice [52]. In addition to saponins, phenolics and flavonoids are also active ingredients in the crude extract of *C. oleifera* tea that have the potential to present interesting biological activities in 5–FU-challenged HaCaT cells. For instance, hyaluronic acid–phenylboronic acid–tea polyphenol nanoparticle hydrogels exhibited efficient antioxidant, ROS-scavenging, and anti-inflammatory effects in oxidative HaCaT cells, as well as wound-healing and tissue-remodeling properties in diabetic mice [53]. In addition, green tea epicatechin–3–gallate mitigated hydrogen peroxide production and the death activation of extracellular signal-regulated kinase of ultraviolet-induced HaCat cells in a concentration-dependent manner [54,55]. Likewise, green tea epigallocatechin–3–gallate could prevent tachyphylaxis and inflammation of dexamethasone-induced HaCaT cells by suppressing the expression of the migration inhibitory factor for the following: IL–6, IL–18, TGF–β, and C–C motif ligands 17, 22, and 10 [56]. Moreover, EGCG could prevent ROS-evoked apoptosis of ultraviolet B-exposed HaCaT cells by downregulating caspases 8 and 3 gene expressions [57]. Furthermore, EGCG could reduce the phosphorylation of ERK, p38, and c–jun amino–terminal kinase, as well as suppress cyclooxygenase 2 expressions and prostaglandin production in HaCaT cells, through 2,2′–azobis (2–amidinopropane) dihydrochloride. This is suggestive of its an anti-inflammatory property [58]. Interestingly, plant flavonoids, such as quercetin, can promote anti-inflammatory and antioxidant wound healing, as well as keratinocyte migration in HaCaT cells via ERK1/2 MAPK and NF–κB pathways by suppressing the expression of IL–1β, IL–6, and IL–8, while upregulating the expression of superoxide dismutase, catalase, GPX, and IL–10 [59,60].

In terms of the study’s limitations, we lacked a high-performance liquid chromatography/electrospray ionization–mass spectrometry and ultrahigh-performance liquid chromatography/electrospray ionization–quadrupole time-of-flight/mass spectrometry machine to identify other active phytochemicals in our crude tea saponins. In addition, the effects of tea saponins on levels of IL–6 and TNF–α, and other cytokines, such as IL–4, IL–13, IL–22, TNF–β, and interferons α and β, that directly target keratinocytes, have not been investigated due to their high cost of analysis. Moreover, the cell signaling pathways, gene expression regulation, and synergistic effects of active ingredients have not been investigated because of insufficient laboratory facilities.

## 5. Conclusions

Tea saponin exhibited potent protective effects against 5–FU-induced ferroptosis, oxidative stress, inflammation, and impaired wound healing. Accordingly, the compound mixture could inhibit lipid peroxidation, restore glutathione peroxidase–4 activity and glutathione content, relieve inflammation, and enhance the repairing of skin keratinocyte injuries that had been induced by 5–FU. These findings highlight the potential of camelliasaponins, metesaponin, and assamsaponin, as well as phenolics and flavonoids in tea seeds as an adjuvant to counteract the deleterious effects of chemotherapy in dentistry, paving the way for clinical investigations. Thus, future studies should continue to explore the specific molecular mechanisms of tea saponin’s active ingredients, verify the efficacy and safety of used regimens, and optimize their application protocols to provide for more efficient large-scale clinical trials.

## Figures and Tables

**Figure 1 nutrients-17-00764-f001:**
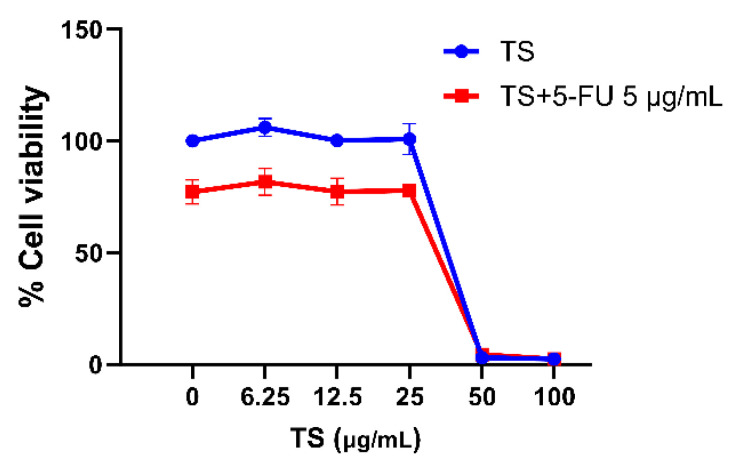
Dose responses of TS on viability of HaCaT cells with and without 5–FU treatment. Data are expressed as mean ± SD values of three separate experiments.

**Figure 2 nutrients-17-00764-f002:**
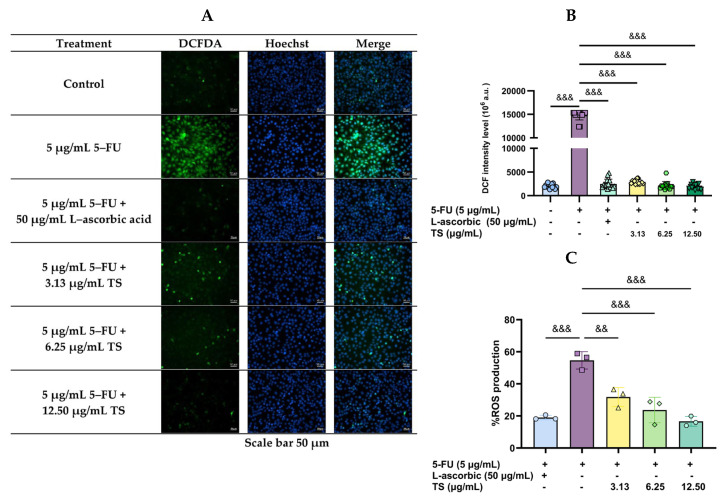
Levels of ROS in HaCat cells were induced by 5–FU and followed by treatment with TS or L–ascorbic acid. Results are shown in fluorescence cell imaging (**A**) and presented as mean ± SD values of three separate experiments (**B**,**C**). Accordingly, ^&&^
*p* < 0.01, ^&&&^
*p* < 0.005 when compared without TS treatment.

**Figure 3 nutrients-17-00764-f003:**
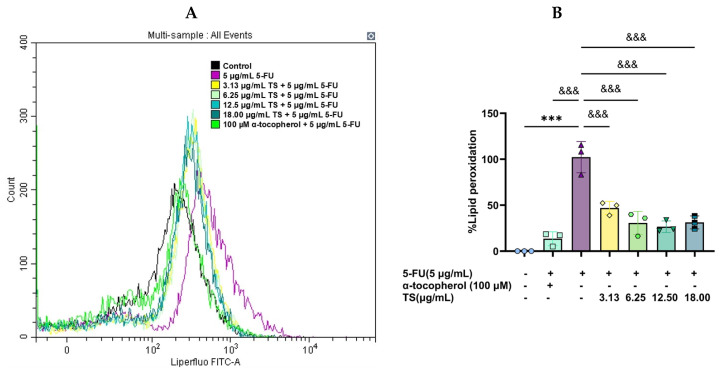
Levels of LPOs in HaCat cells were induced by 5–FU and followed by treatment with TS or α–tocopherol. Results are shown in histograms (**A**) and presented as mean ± SD values of three separate experiments (**B**). Accordingly, *** *p* < 0.005 when compared without the 5–FU induction, ^&&&^
*p* < 0.005 when compared without TS treatment.

**Figure 4 nutrients-17-00764-f004:**
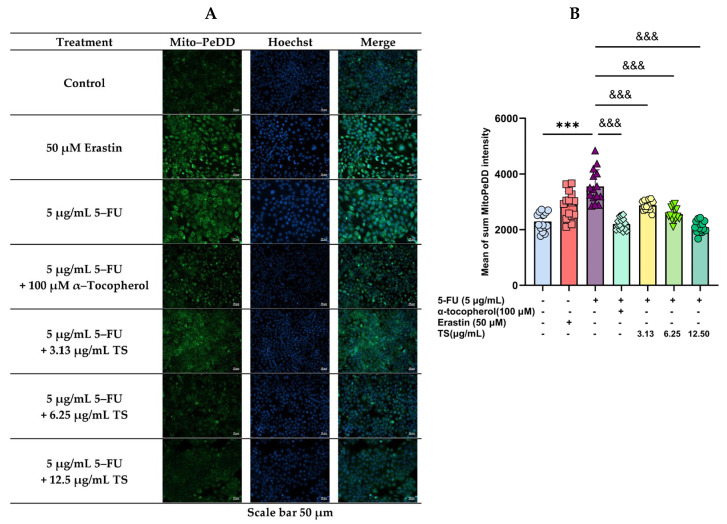
Mitochondrial redox-active iron levels in HaCat cells were induced by 5–FU and followed by treatments with TS, α–tocopherol, or erastin. They were then identified using confocal fluorescence microscopy. Results are illustrated in fluorescence cell imaging (**A**) and presented as mean ± SD values of three separate experiments (**B**). Accordingly, *** *p* < 0.005 when compared without 5–FU induction; ^&&&^
*p* < 0.005 when compared without the compound treatments.

**Figure 5 nutrients-17-00764-f005:**
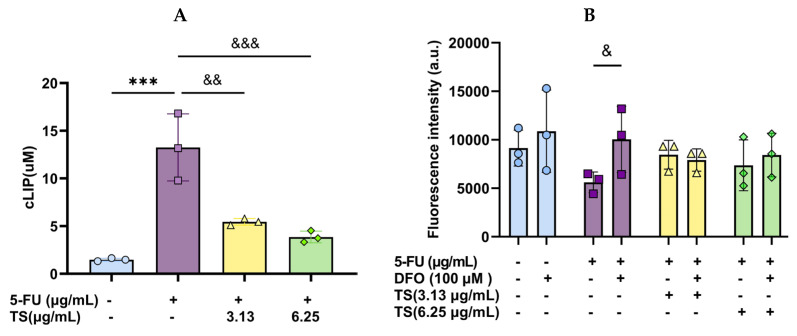
Cytosolic LIP levels in HaCat cells were induced by 5–FU and followed by TS treatment without/with DFO chelation. Data obtained from three separate experiments of monotherapy (**A**) and combined therapy (**B**) are expressed as mean ± SD values. Accordingly, *** *p* < 0.005 when compared with control cells; ^&^
*p* < 0.05, ^&&^
*p* < 0.01, and ^&&&^
*p* < 0.005 when compared with 5–FU induction alone.

**Figure 6 nutrients-17-00764-f006:**
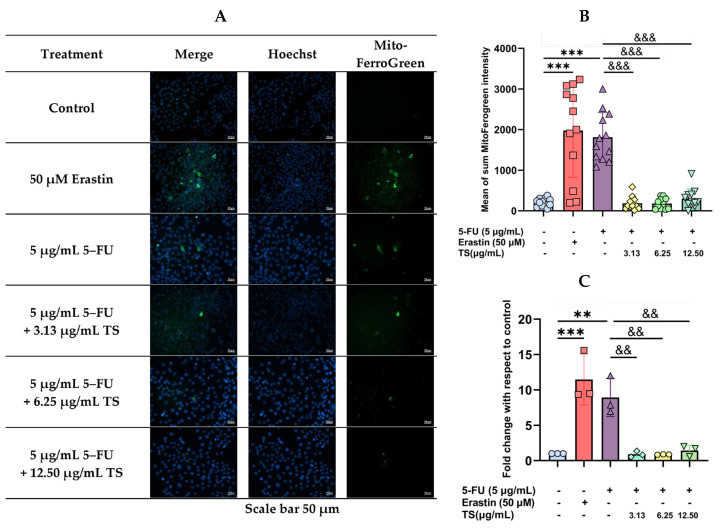
Mitochondrial LIP levels in HaCat cells were induced by erastin and 5–FU followed by TS treatments. Results were expressed in fluorescence cell imaging (**A**) and data obtained from three separate experiments (**B**,**C**) and are expressed as mean ± SD values. Accordingly, ** *p* < 0.01, *** *p* < 0.005 when compared with control cells; ^&&^
*p* < 0.01, and ^&&&^
*p* < 0.005 when compared with 5–FU induction alone.

**Figure 7 nutrients-17-00764-f007:**
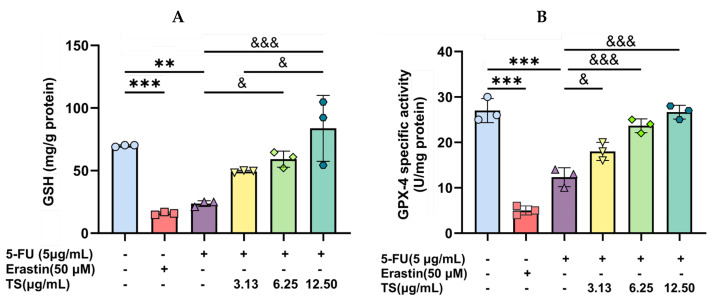
Levels of GSH- (**A**) and GPX–4-specific activity (**B**) in HaCat were cells depressed by 5–FU or erastin and followed by TS treatment. Data are expressed as mean ± SD values of three separate experiments. Accordingly, ** *p* < 0.01, *** *p* < 0.005 when compared without 5–FU induction; ^&^
*p* < 0.05, and ^&&&^
*p* < 0.005 when compared without compound treatments.

**Figure 8 nutrients-17-00764-f008:**
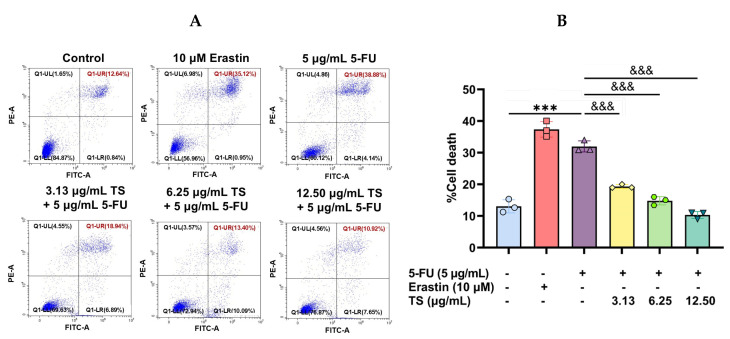
Viability of HaCaT cells were induced by erastin or 5–FU followed by treatments of TS. The percentage of Fluor™488/DAPI-labeled apoptotic cells are shown in scatter plots (**A**) and presented as mean ± SD values of three separate experiments (**B**). Accordingly, *** *p* < 0.005 when compared without 5–FU induction; ^&&&^ *p* < 0.005 when compared without compound treatments.

**Figure 9 nutrients-17-00764-f009:**
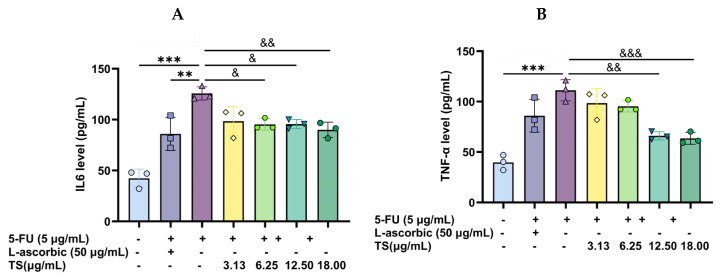
Production of IL–6 (**A**) and TNF–α (**B**) from HaCat cells induced by 5–FU and treated with TS. Data obtained from three separate experiments are expressed as mean ± SD values, which ** *p* < 0.01 and *** *p* < 0.005 when compared without 5–FU induction; ^&^
*p* < 0.05, ^&&^
*p* < 0.01, and ^&&&^
*p* < 0.005 when compared with compound treatments.

**Figure 10 nutrients-17-00764-f010:**
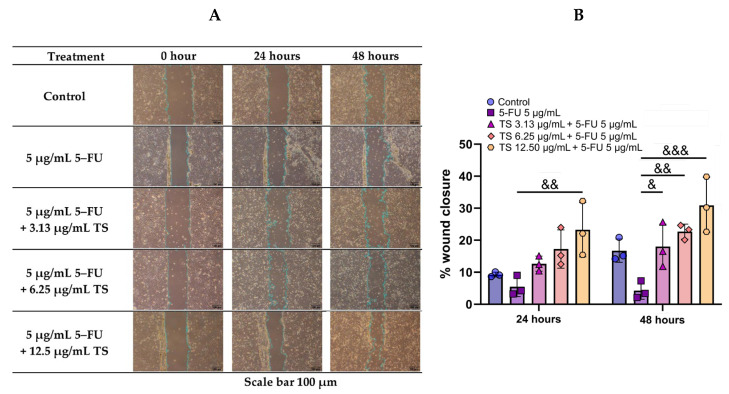
Effect of TS treatment on 5–FU-induced HaCaT cell wound healing. Results are shown in microscopic cells (**A**) and mean ± SD values of three separate experiments (**B**). Accordingly, ^&^
*p* < 0.05, ^&&^
*p* < 0.01 and ^&&&^
*p* < 0.005 when compared without compound treatments.

## Data Availability

Datasets are available from the authors upon request.

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
