# Peer review of "Camellia Tea Saponin Ameliorates 5–Fluorouracil-Induced Damage of HaCaT Cells by Regulating Ferroptosis and Inflammation"

_nutrients, 2025, doi:10.3390/nu17050764_

Round 1

Reviewer 1 Report

Comments and Suggestions for Authors

This is an interesting paper presenting the benefits of tea saponin extract in countering some side effects of cancer-treatment medications. It is adequate for the topic of the special issue of Nutrients that the authors submitted it to.

Some issues that need to be resolved before publication are listed below:

Line 47: unjustified font increase for reference citations

Line 56: saponins (plural)

Last paragraph of the Introduction - there is a very sudden switch from Anemoside and Astragaloside to tea saponin, with no transition at all. Existing background literature should be better emphasized and used to turn this sudden switch to a smoother one. What are some specific phytochemicals in Tea saponin mixture according to available literature that motivated this very targeted study?

Line 108: Have you performed the HPLC quantitative analysis yourselves, in your laboratory? In order to be able to report gram amounts as you do, analytical standards (pure compounds) should be available to calibrate the HPLC detector and attribute peak areas to gram quantities. Did you have such analytical standards available and have you made use of them? (provide calibration equations in this case)

Line 117: all technical details about the CO2 incubator used must be provided

Line 154: all technical details about the centrifuge must be added (type, manufacturer, place, country)

Line 293: What is your rationale for reporting sometimes standard deviations and other times standard errors of the means?

Figures 2, 3, 4, 6, 8, 10: font sizes are often too small to allow easy reading of values

Lines 447-479: all this information belongs to the Introduction, not the Discussion part of the manuscript

Line 568: not compound, but "compound mixture", b/c "tea saponin" is a generic term for a mixture of phytochemicals and you have not identified what is the specific compound or compound class responsible for the effects you observed in your work

References - some of the references are missing article number or page number (e.g. references no. 14, 16, 19, 27, 42, 43)

Author Response

Best regards,

Professor Somdet Srichairatanakool, PhD.

Reviewer 2 Report

Comments and Suggestions for Authors

The manuscript presents an interesting study of the effect of tea saponin extract on 5-FU treated HaCaT cells as a model of keratinocytes. The message of the study is not clear. Several effects of the extract were documented but they did not affect the survival of the cells at levels relevant to the chemotherapeutic effect. The protective effect concerns only lower concentrations of the extract when most of the cells remain viable.

The effects observed are due to the tea saponin extract. Its rough composition is reported; it contains also other biologically active compounds like phenolics and, among them, flavonoids. This could be mentioned in the discussion; in the future, perhaps saponins could be separated from other compounds to exclude the possible effects of the latter.

The study is properly done and presented. The methods are scrutinously described.

Lines 25/26: “exerted epithelial repairing effects against skin injuries”. This is an overstatement. The authors did not study skin injury, they used only the wound healing assay which is an in vitro model. The results of the study point to but do not prove such a possibility.

Line 27: “tea saponin”, more precisely: “tea saponin extract”

Lines 60/61: “decrease mitochondrial reactive oxygen 60 species (ROS). It can also decrease glycogen synthase kinase 3beta”, better: “decrease the level of…”

Lines 145/146:” Cytosolic LPOs”, Cytosol is defined as a solution surrounded and separated by membranes; as lipid peroxidation is confined to membranes, the term” cytosolic LPO” does not seem appropriate.

Did the authors determine IC50 of HaCaT  cells in the presence of TS?

Line 382: The authors should cite Figure 7A and 7B instead of 6A and 6B.

Lines 387/388: ”Levels of GSH (A) and GPX–4 specific activity (B) in HaCat were cells induced by 5–FU or erastin”,please correct this phrase, the levels were depressed and not induced by 5-FU 

Line 404: “Viability of HaCaT cells were induced by erastin or 5–FU”, please correct this statement.

References: journal abbreviations should contain periods.

Although the English of the text is generally not bad, an inspection by a native speaker should help in the optimization of some statements, especially in legends to Figures 2-8. In some places, capital letters are used without reason.

Comments on the Quality of English Language

Although the English of the text is generally not bad, an inspection by a native speaker should help in the optimization of some statements, especially in legends to Figures 2-8. In some places, capital letters are used without reason.

Author Response

Yours sincerely,

Professor Somdet Srichairatanakool, PhD.

Reviewer 3 Report

Comments and Suggestions for Authors

The manuscript presents an innovative review of the use and protective role of TS in mitigating 5–FU–induced ferroptosis and inflammation in human keratinocytes. The authors thoroughly investigate and evaluate the effect of the protector on immune infiltration. The formation, development and progression of 5–FU–induced ferroptosis and inflammation in human keratinocytes; and the additional inflammatory microenvironment of this disease are examined in depth. The authors emphasize the fact that TS mitigating 5–FU–induced ferroptosis, reduces symptoms, suggesting that the use of TS protects and stops chemotherapy–induced mucositis.
The manuscript meets the requirements of the journal, is written in standard English and presents in-depth innovative scientific information in the field. In addition, the authors thoroughly analyze the role of the protector in promoting the prevention of inflammation. In addition, innovative methods were used to support the relevance of the thesis.
The presented introduction and subsections, materials and methods and conclusions are satisfactory. There are some grammatical and stylistic inaccuracies that should be corrected. The figures cited are of good resolution, well described and structured.
The limitations of the study are not clear. The conclusion is satisfactory. The cited literature is appropriate and comprehensive. To strengthen the discussion, introducing a scheme is appropriate.

Comments on the Quality of English Language

-

Author Response

(The authors gave the same response as above.)
